# Effect of health education in the control of schistosomiasis in Dass Emirate Council of Bauchi State, Nigeria: An intervention study

**Sunday Charles Adeyemo**[1]*, **Gbadebo Jimoh Oyedeji**[2], **James Atolagbe**[3], **Oladunni Opeyemi**[3], **Sunday Olarewaju**[4], **Calistus Akinleye**[5], **Funso Olagunju**[5], **Eniola Dorcas Olabode**[6], **Abdulwaris Salisu Maleka**[7]

1 Institut Superieur de Sante, Niamey, Niger Republic, 2 Department of Medical Microbiology and Parasitology, LAUTECH, Osogbo, Nigeria, 3 Public Health Department, Adeleke University, Ede, Nigeria, 4 Community Medicine Department, Osun State University, Osogbo, Nigeria, 5 Department of Paediatric Medicine, Osun State University, Osogbo, Nigeria, 6 Department of Community Health, Obafemi Awolowo University, Ile-Ife, Nigeria, 7 Disease control and immunization unit, Bauchi State Primary Health Care Development Agency, Dass Emirate, Nigeria

* charlespatho@gmail.com

**Data Availability Statement:** All data are in the manuscript and/or supporting information files.

**Funding:** The author(s) received no specific funding for this work.

## Abstract

### Objective

Schistosomiasis has been recognized by WHO as a major contributor to mortality and morbidity, particularly in Sub-Sahara Africa, where it is most prevalent. There is a lack of reliable data on the effectiveness of health education interventions in reducing the prevalence of schistosomiasis in Bauchi State. Hence, the study assessed the prevalence of schistosomiasis and the knowledge, attitude and practices of community members of Dass Emirate towards the prevention and control of schistosomiasis before and after health education intervention.

### Results

At pre-intervention, the majority of the respondents 234 (66.9%) have been diagnosed, or have family members or community members who have been diagnosed with schistosomiasis. Ninety-six (27.5%) of respondents have good knowledge about schistosomiasis. 79 (22.6%) of the respondents strongly agree that they can confidently recognize symptoms of schistosomiasis. Only 91 (26.0%) strongly agreed to taking responsibilities for taking preventive measures. At post-intervention, the prevalence of schistosomiasis dropped to 55.1%. This was statistically significant at p = 0.043 using McNemar's test as a test of significance. Knowledge about schistosomiasis increased from 27.5% to 87.0% at post-intervention. This was statistically significant at p <0.05. Regarding attitudes and practices, good attitudes and practices increased from 59.1% at pre-intervention 71.0% at post-intervention. However, this was not statistically significant (p>0.05). Health education and education level of respondents were predictors of Knowledge, Attitude and Practices scores.

**Competing interests:** The authors have declared that no competing interests exist.

## Author summary

Schistosomiasis is a major public health concern in many tropical and subtropical regions, affecting millions worldwide. When human come in contact with the larvae of the parasitic trematode called *Schistosoma* through contaminated fresh water, the larvae penetrate human skin and mature in the blood vessels, leading to various clinical manifestations. The disease can lead to serious health complications, including liver and spleen enlargement, kidney damage, bladder cancer, and even death if left untreated. The primary strategy towards control of schistosomiasis was mass drug administration (MDA) using praziquantel, targeting school-aged children and at-risk communities. However, despite these efforts, schistosomiasis persisted, primarily due to challenges such as inadequate drug supplies, infrastructural deficits, and socio-cultural barriers. The emphasis then shifted from mere disease control to mitigating it as a public health issue, therefore the authors introduced a health education intervention in order to improve the knowledge, attitude and practices of the community members on the prevention and control of schistosomiasis. The findings of this study has implications on mitigating schistosomiasis, and contributing to the global body of knowledge on practical approaches to controlling schistosomiasis and guide the development of evidence-based strategies that can reduce the burden of the disease worldwide.

## Introduction

Schistosomiasis, commonly known as bilharzia, is a parasitic disease caused by trematode flatworms of the genus *Schistosoma*. It is a major public health concern in many tropical and subtropical regions, affecting millions worldwide. The disease is transmitted through contact with fresh water contaminated with parasitic larvae, which penetrate human skin and mature in the blood vessels, leading to various clinical manifestations [1]. Globally, schistosomiasis is recognized as a significant contributor to morbidity and mortality, especially in sub-Saharan Africa. The World Health Organization (WHO) estimates that over 200 million people are infected with schistosomiasis worldwide, with most cases occurring in Africa [2]. The disease can lead to severe health complications, including liver and spleen enlargement, kidney damage, bladder cancer, and even death if left untreated [3].

In a study on the challenges of controlling schistosomiasis through mass drug administration (MDA) campaigns were highlighted. While MDA has shown short-term benefits in reducing the prevalence and intensity of infection, the long-term impact remains uncertain [4]. The study emphasized the need for comprehensive strategies, including health education and improved sanitation, to achieve sustainable disease control. There is a lack of reliable data on the effectiveness of health education interventions in reducing the prevalence of schistosomiasis in Bauchi State. This knowledge gap makes it difficult to design effective control programs and allocate resources efficiently. Therefore, this study aims to evaluate the effectiveness of health education as an intervention in controlling schistosomiasis in Dass Emirate Council of Bauchi State.

## Methods

### Ethics statement

Ethical approval was obtained from Research Ethics Committee of Ministry of Health, Bauchi State with the registration number, BSMOH/REG/81/2023. A written formal consent was also

obtained from participants after providing them with detailed information about the study's purpose, the procedures involved, the potential risks and benefits, and their rights as participants. A written formal consent was also obtained from the parents of the participant who were less than 18 years of age. Confidentiality was strictly maintained throughout the study. All participant data were anonymized and stored securely, accessible only to the research team. Identifiers were removed or coded to prevent the identification of individual participants in any reports or publications resulting from the research.

The study employed a time-series quasi-experimental design which involved pre- and post-intervention data collection. The intervention involved structured health education programs and campaigns in the communities. A total of 17 Research Assistants who served as enumerators and health educators were recruited. The research assistants were dwellers of the communities who can speak and understand Hausa language. This is to ensure proper communication between them and the community members. 15 of the research assistants were health care workers with the primary health care board in Bauchi State, who helped in engaging the community members at the community health centers. Two were students of Bauchi State University studying Statistics. The head of the health care workers have Masters in Health Care Economics while others have national diploma (ND) and higher national diploma (HND) in Health education, Health information technology and community health from School of Health. This aided adequate understanding of the research work and communication between them and the researcher, especially during the training.

The health education was conducted by the trained health educators for four hours per contact in all the ten communities selected in Dass Emirate. A training targeted population at risk and major stakeholders including, school pupils, market people, religious leaders, farmers and herders in 10 communities of Dass Emirate of Bauchi where schistosomiasis is mostly prevalent. The duration of the health education was 4 weeks due to the nature of occupation of the respondents who were predominantly farmers and fisher men who needed to go to their various farms and the insecurity issues peculiar with the study location.

The training manual of the Federal Ministry of Health of Nigeria which contained information on the identification of schistosomiasis symptoms, the mode of transmission as well as preventive measures of schistosomiasis was adapted during the health education. The health educators ensured the health education sessions were interactive in order to understand the perception and attitude of the community members towards schistosomiasis. They further emphasized the need for a change in behavior in order to promote the prevention of schistosomiasis. The strategies of the health education campaign involved community participation by creating a schistosomiasis booth at the community field.

Another strategy was peer education which involves identification of champions of the group which included community leaders, religious leaders, district heads, youth leaders, market women and leaders and principals of Almanjiri schools who served as peer educators to increase awareness about the prevention and control of schistosomiasis in the communities. Also, community outreach was done to hard-to-reach. There was also sharing of leaflets and pasting of posters containing information on the prevention and control of schistosomiasis in the communities.

Self-administered semi-structured questionnaires were used to collect baseline data from the participants before the intervention. This was done after the questionnaire was piloted in a similar setting to the study area to refine questions, ensure cultural appropriateness, and assess the clarity of the language used. Feedback from the pilot was used to make necessary adjustments to the data collection instruments. After the intervention, the same questionnaire used to collect baseline data was used to collect post-intervention data from the participants. In order to avoid cognitive bias, singular blinding was done during the pre- and post-intervention data collection whereby the respondents were not aware of the purpose of data collection.

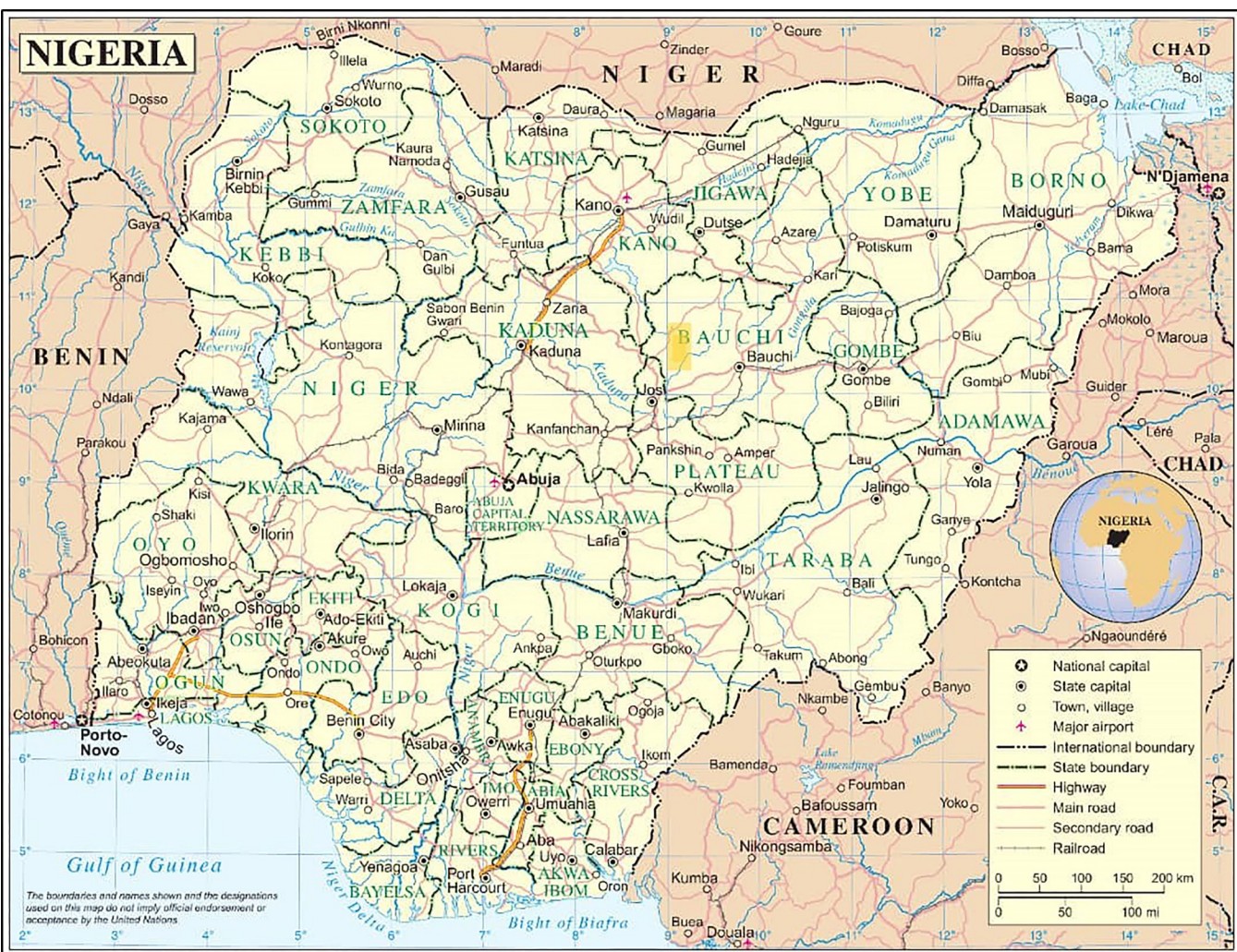

**Fig 1. Map of Nigeria showing Bauchi State and Dass Emirate [18].** Available at http://www.un.org/Depts/Cartographic/map/profile/nigeria.pdf.

The study population of this study encompassed residents of the Dass Emirate Council (depicted with yellow box in Fig 1) of Bauchi State. Several factors underpin the high prevalence in Bauchi State. The state's geography, characterized by numerous freshwater bodies, which may be susceptible to contamination. The demographic composition of the Dass Emirate Council, as per the 2006 census, indicates a population that is youthful with a median age significantly lower than the national average. This demographic factor is pivotal in understanding the transmission dynamics of schistosomiasis, as younger populations often exhibit higher exposure to infection due to their recreational activities in infested water bodies. (Federal Republic of Nigeria Official Gazette, 2007).

The Fisher's formula ($n = z^2pq/d^2$) was used to calculate the sample size (where z = 1.96 at 95% confidence interval, p = 60% from the study of Sacolo et al., 2018, q = 1-p, d = 0.05). The result was approximated to 450 after calculating for non-response. In order to avoid selection bias, multistage sampling technique was used to recruit respondents.

**Stage 1**: 10 communities were randomly selected from the communities in Dass emirate. Proportional allocation was used to determine the number to be selected from each community.

**Stage 2**: Stratified sampling technique as used in the study of [5] was used to stratify the population in each community into 3 strata based on their age group. The strata were 15–24 years, 25–64 years and 65 years and above. The number to be selected from each stratum was calculated by dividing the number of persons in each group by the total population of the community, multiplied by the total number to be selected in the community.

**Stage 3**: The respondents were selected from each stratum using simple random sampling technique.

The data collected were analyzed with SPSS software package version 25, and a p- value of less than 0.05 was considered statistically significant. Descriptive statistics were done for all variables. Comparison of variables at pre-and post-intervention was done using McNemar's test as a test of association. Regression analysis was used to identify predictors of Knowledge, Attitude and Practices (KAP) scores.

## Results

There are 148 (42.3%) of the respondents were within age range 15–24 while 155 (44.3%) of the respondents were within the age range 25–64 years and 47 (13.4%) were within the age range 65 and above. Majority of the respondents 195 (55.7%) were male while 155 (44.3%) were female. Sixty-five (18.6%) of the respondents had no formal education, 45 (12.9%) had primary education while (128, 36.5%) and (112, 32.0%) had secondary and tertiary education respectively. Less than half of the respondents, 87 (24.9%) were students, 127 (36.3%) were farmers, 64 (18.3%) were fishers, 56 (16.0%) were traders, and 16 (4.6%) were civil servants. Majority of the respondents, 220 (62.9%) were Muslims. Majority of the respondents 217 (62.0%) were married, 104 (29.7%) were single, 7 (2.0%) were divorced, and 18 (5.1%) participants. More than half 296 (84.6%) of the respondents were rural dwellers while 54 (15.4%) were urban dwellers. Majority of the respondents had lived in Dass Emirate Council for more than 10 years 263 (75.1%), 23 (6.6%) had lived there for less than a year, 20 (5.7%) had lived there for 1–5 years while 36 (10.3%) for 6–10 years.

At pre-intervention, the majority of the respondents 234 (66.9%) have been diagnosed, or had family members or community members who have been diagnosed with schistosomiasis. Ninety-five (27.1%) reported that schistosomiasis is very common in their community, 131 (37.5%) reported that schistosomiasis is somewhat common in their community, 43 (12.3%) reported that schistosomiasis is rare in their community while 81 (23.1%) are unsure about the commonness of schistosomiasis in their community. More than half, (207, 59.1%) reported that someone in their community had been diagnosed with schistosomiasis.

About knowledge of schistosomiasis at pre-intervention, overall good knowledge was 27.5%. The majority (310, 88.6%) of the respondents have heard about schistosomiasis. Majority of the respondents 200 (64.5%) heard about schistosomiasis from healthcare workers. One hundred and twenty-eight (41.3%) of the respondents reported abdominal pain as a common symptom of schistosomiasis, 36 (11.6%) respondents reported blood in urine or stool as a common symptom of schistosomiasis, 71 (22.9%) reported fatigue as a common symptom of schistosomiasis, while 85 (27.4%) respondents reported weight loss as a common symptom of schistosomiasis. About transmission of schistosomiasis, 83 (26.8%) reported that it can be transmitted through infected water sources, 140 (45.2%) reported that it can be transmitted through mosquito bites, 75 (24.2%) reported that it can be transmitted through contaminated food while 52 (16.8%) reported that it can be transmitted through human contact. Ninety-one (29.4%) reported that schistosomiasis is preventable, while 140 (45.2%) reported that it is not preventable and 79 (25.4%) do not know. On the preventive measures of schistosomiasis, out of 91 that said it can be prevented, 20 (30.0%) reported that it can be prevented by avoiding

contact with contaminated water, 31 (34.1%) reported that it can be prevented by boiling or treating water before use, 29 (31.9%) reported that it can be prevented by proper sanitation and hygiene practices while 10 (11.0%) reported that it could be prevented by use of insecticide-treated nets.

About attitude and practices towards the prevention of schistosomiasis, at pre-intervention, 79 (22.6%) of the respondents strongly agree that they can confidently recognize symptoms of schistosomiasis. Fifty-five (15.7%) strongly agree that there is community engagement in efforts towards prevention of schistosomiasis. Only 91 (26.0%) strongly agreed to take responsibility for taking the preventive measures and only 23 (6.6%) strongly agreed that they discuss prevention of schistosomiasis with their family. However, 118 (33.7%) agreed that their actions can reduce the risk of schistosomiasis. One hundred and twenty (34.3%) avoided contact with contaminated water to prevent and control schistosomiasis, 51 (14.6%) boiled or treated water to prevent and control schistosomiasis, 129 (36.8%) practiced proper hygiene and sanitation to prevent and control schistosomiasis while 50 (14.3%) used insect treated nets to prevent and control schistosomiasis.

At post-intervention, the prevalence reduced to 190 (55.1%) and this was statistically significant at p = 0.043. (Fig 2) Knowledge of schistosomiasis increased to 87.0% post-intervention, which was statistically significant at p <0.001. (Fig 3) About attitude and practices towards schistosomiasis prevention and control, 126 (36.5%) of the respondents strongly agree that they can confidently recognize symptoms of schistosomiasis. This difference was statistically significant at p = 0.045. Eighty-eight (25.5%) strongly agree that there is community engagement in efforts towards prevention of schistosomiasis. However, this difference was not statistically significant (p>0.05). One hundred and eighteen (34.2%) strongly agreed to taking responsibilities for taking preventive measure. The difference is statistically significant at p = 0.035. One hundred and fifty-eight (45.8%) agreed that they discuss prevention of schistosomiasis with their family. This difference was statistically significant at p = 0.022. One

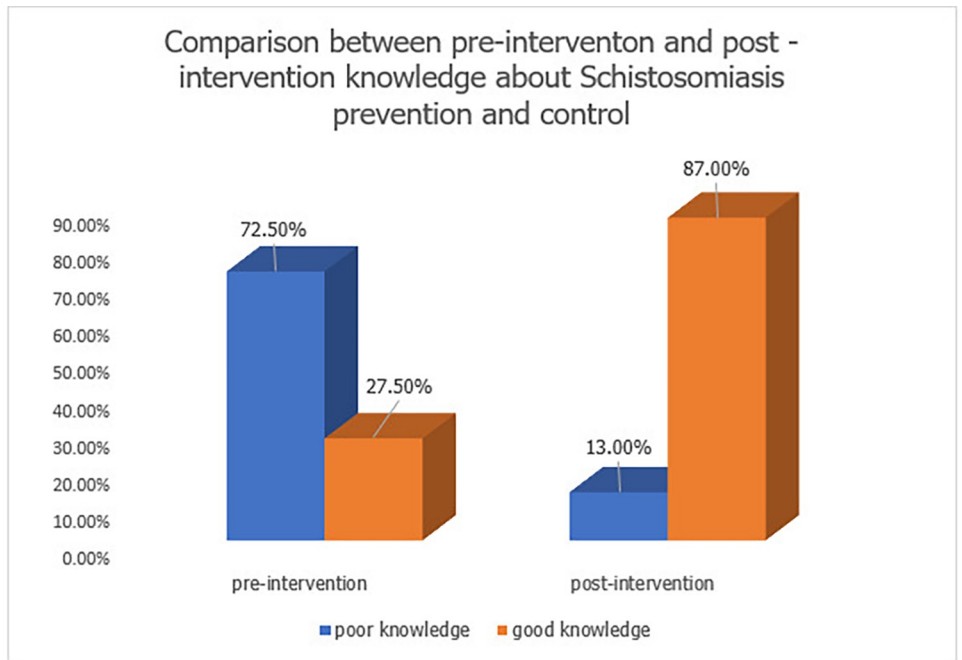

**Fig 2. Comparison between pre-intervention and post-intervention knowledge about schistosomiasis prevention and control.**

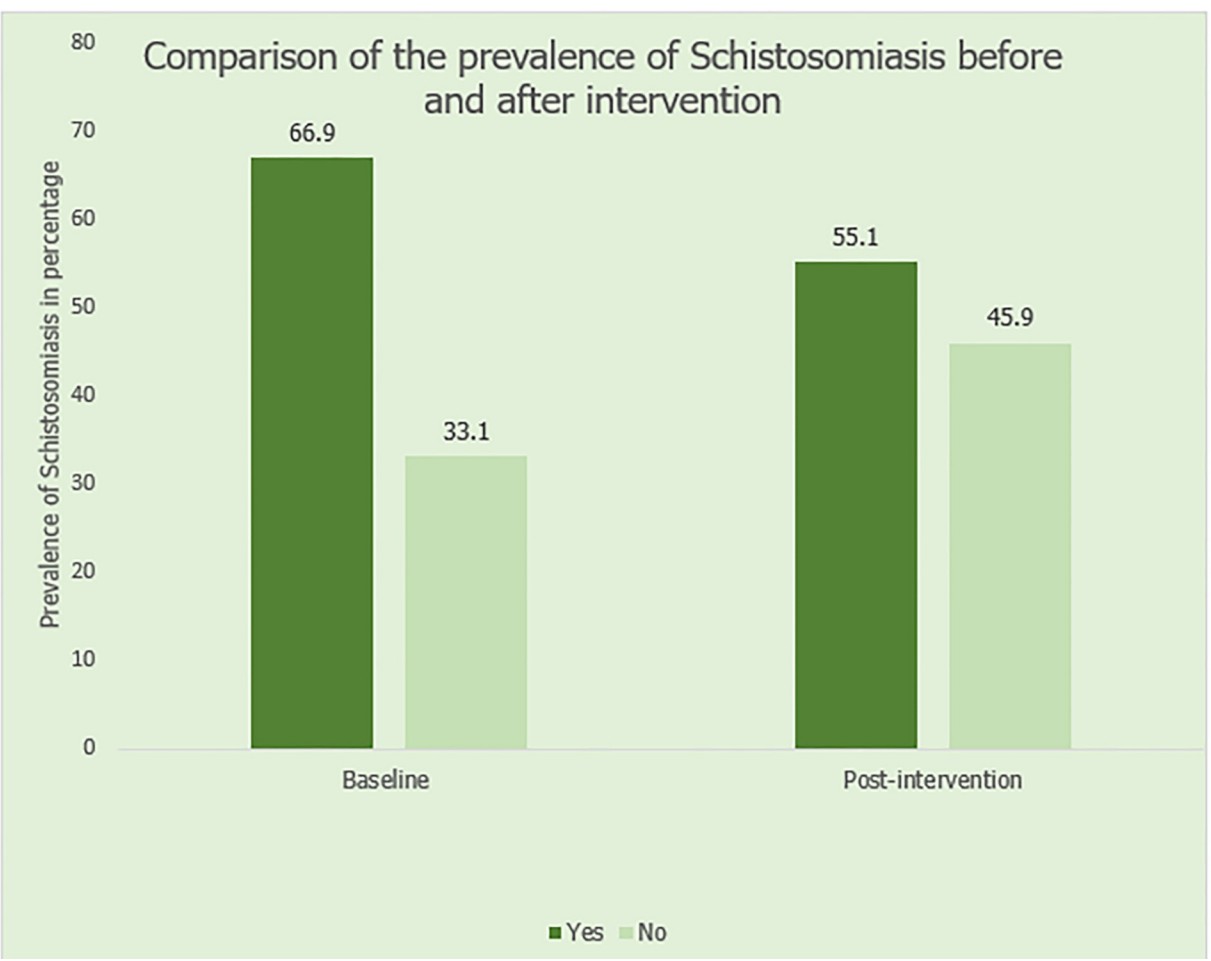

**Fig 3. Comparison of the prevalence of schistosomiasis before and after intervention.**

hundred and twenty-four (35.9%) respondents strongly agreed that their actions can reduce the risk of schistosomiasis. However, this was not statistically significant (p>0.05). About actions taken to prevent schistosomiasis, 120 (34.8%) reported avoiding contact with contaminated water, 51 (14.8%) reported boiling or treating water before use, 124 (35.9%) reported proper sanitation and hygiene practices and 50 (14.5%) reported use of insecticide-treated nets. However, this difference was not statistically significant (p>0.05) (Table 1).

Using regression analysis to identify the predictors of KAP scores, Health Education Score was a strong positive predictor of KAP scores (β = 0.479, p <0 .001), indicating that higher health education correlates with better knowledge, attitudes, and practices. Among demographic factors, Education Level emerged as a significant predictor of KAP scores (β = 0.090, p = 0.006), while Gender showed a near-significant relationship (β = 0.063, p = 0.055). Occupation and the number of children had no significant predictive power.

## Discussion

The prevalence of schistosomiasis was high at baseline. More than half (66.9%) of the respondents have been diagnosed with schistosomiasis. This can be attributed to the fact that 42.3% of the respondents are adolescents and are prone to activities such as swimming and fishing in dirty water, which can expose them to the disease [6]. Additionally, this high prevalence can be

**Table 1. Comparison of attitude and practices towards schistosomiasis prevention and control among respondents at pre-intervention and post-intervention.**

| Attitude and practices towards Frequency (%) Statistics prevention and control of Baseline Post-intervention | | | |
|---|---|---|---|
| schistosomiasis | (n = 350) | (n = 345) | |
| **I can confidently recognize schistosomiasis symptoms** | | | p = 0.045* |
| Strongly Disagree | 109 (31.1) | 50 (20.4) | |
| Disagree | 71(20.3) | 51 (14.8) | |
| Neutral | 55 (15.7) | 49 (14.3) | |
| Agree | 36 (10.3) | 69 (20.0) | |
| Strongly agree | 79 (22.6) | 126 (36.5) | |
| **There is community engagement in prevention efforts** | | | p = 0.058 |
| Strongly disagree | 120 (34.3) | 59 (17.1) | |
| Disagree | 91 (26.0) | 64 (18.6) | |
| Neutral | 49 (14.0) | 48 (13.9) | |
| Agree | 35 (10.0) | 86 (24.9) | |
| Strongly agree | 55 (15.7) | 88 (25.5) | |
| **I take responsibility for taking preventive measures** | | | p = 0.035* |
| Strongly disagree | 108 (30.9) | 20 (5.8) | |
| Disagree | 55 (15.7) | 56 (16.2) | |
| Neutral | 37 (10.5) | 47 (13.6) | |
| Agree | 59 (16.9) | 104 (30.1) | |
| Strongly agree | 91 (26.0) | 118 (34.2) | |
| **I have discussions about prevention with family** Strongly disagree | | | p = 0.022* |
| Disagree | 141 (40.2) | 54 (15.7) | |
| Neutral | 70 (20.0) | 52 (15.1) | |
| Agree | 65 (18.6) | 17 (4.9) | |
| Strongly agree | 51 (14.6) | 158 (45.8) | |
| | 23 (6.6) | 64 (18.6) | |
| **I belief that my actions can reduce risk of schistosomiasis** | | | p = 0.055 |
| Strongly disagree | 90 (25.7) | 51 (14.8) | |
| Disagree | 45 (12.9) | 53 (15.4) | |
| Neutral | 36 (10.3) | 40 (11.6) | |
| Agree | 118(33.7) | 77 (22.3) | |
| Strongly agree | 61 (17.4) | 124 (35.9) | |
| **Actions you/ family members has taken to prevent schistosomiasis** | | | p = 0.075 |
| Avoiding contact with contaminated water Boiling or treating water before use | 80 (22.9) | 120 (34.8) | |
| Proper sanitation and hygiene practices | 81 (23.1) | 51 (14.8) | |
| Use of insecticide-treated nets | 89 (25.4) | 124 (35.9) | |
| | 100 (28.6) | 50 (14.5) | |

$\chi^2$ = Chi-square, df = degree of freedom, p = p-value using McNemar test, * = statistically significant

attributed to the fact that 84.6% are rural dwellers which are prone to poor sanitation and hygiene practices due to unavailability of infrastructures for proper sewage disposal treatment and poor access to clean water as described in WHO newsletter on schistosomiasis. [7,8] This is similar to a study who reported that children and adolescents are more prone to schistosomiasis and poor sanitation is also linked with the disease [6].

The Knowledge regarding schistosomiasis was poor at baseline. Only 33.1% had overall good knowledge about schistosomiasis. The poor knowledge can be attributed to low level of education among respondents. This is similar to the findings on the knowledge attitude and practices regarding rural communities in Kano State [9].

The attitude and practices towards schistosomiasis prevention and control was poor at baseline. More than half (60.3%) of the respondents reported that there was no community engagement in efforts towards prevention of schistosomiasis. However, 33.7% of the respondents agreed that some actions can reduce the risk of schistosomiasis. In order to prevent and control schistosomiasis, 34.3% and 36.8% avoided contact with contaminated water and practiced proper hygiene and sanitation respectively. This reveals significant shifts in the

community's perceptions and experiences with the disease, which is an important factor in achieving behavioral change [10].

At post-intervention, the prevalence of schistosomiasis decreased significantly to slightly more than half, which was statistically significant at p = 0.043 using McNemar's test. This could be attributed to heightened community awareness of schistosomiasis as a result of health education. The reduction is of public health relevance as it serves as an important measure in combating schistosomiasis in other endemic areas which aligns with research that suggests heightened awareness through health education can increase community concern thereby reducing prevalence of schistosomiasis [11].

At post-intervention, overall knowledge improved from 33.1% at baseline to 81.0% after health education intervention. This aligns with the findings which reported that improved symptom recognition was linked to effective health education, potentially leading to earlier disease detection and treatment [12]. The overall attitude and practices concerning schistosomiasis control improved after the health education intervention, however, some of the components were not statistically significant and this could be due to the short duration of the health education intervention. One hundred and eighteen (34.2%) of the respondents strongly agreed to taking responsibilities for taking preventive measures while 18.6% of the respondents agreed that they discuss prevention of schistosomiasis with their family. A study by [13] supports this finding, suggesting that enhanced community engagement is often observed following targeted health education campaigns, which empower communities to take an active role in disease prevention. The role of family discussion in promoting health practices has also been well-documented [11], and the result of this study echo the potential of family-based interventions to amplify health messages within a community. One hundred and twenty-four (35.9%) strongly agreed that some actions can reduce the risk of schistosomiasis. This is indicative of an internalization of health education messages, where community members not only learn about preventive measures but also believe in their efficacy. This internal belief system is essential for sustained behavior change and has been cited in previous studies as a critical component of health education success [14]. The increase in the items indicating Knowledge, Attitude and Practices may be attributed to an enhancement in disease detection and diagnosis efforts, a finding consistent with studies that demonstrate improved identification of endemic diseases following concerted health interventions [15]. Similar patterns have also been noted in other literature, where increased awareness following educational interventions correlates with improved health behaviors [16].

An essential aspect of the study's success was its culturally sensitive approach. By tailoring the intervention to fit the local cultural context and incorporating local beliefs and practices, the program achieved higher engagement and effectiveness. This approach is in line with the recommendations of [11,17], who emphasize the importance of cultural sensitivity in health education interventions.

The findings highlight the importance of health education interventions in improving knowledge, attitudes, and practices towards schistosomiasis control. They underscore the need for addressing perceived barriers and emphasize the role of community engagement in schistosomiasis prevention and control. The study recommends prioritizing formulation of comprehensive health policies that address schistosomiasis within the broader context of public health and increased allocation of resources while encouraging partnership from stakeholders and participation from individuals in combatting schistosomiasis.

## Limitation

The limitation of the study is the reliance on self-reported data, which may be subject to recall bias and social desirability bias. Also, the prevalence of schistosomiasis was not clinically

confirmed which could have impact on generalizability of the study findings. The duration of the health education was also limited due to the nature of occupation of the respondents who were farmers who needed to go to their various farms and the insecurity issues peculiar with the study location.

Future research could consider using objective measures, such as medical records, to validate self-reported prevalence of schistosomiasis. Future research should focus on conducting long-term follow-up studies to assess the sustained impact of health education interventions. Evaluating changes in knowledge, attitudes, and practices over an extended period can provide insights into the durability of behavior change.

## Acknowledgments

The authors wish to acknowledge the respondents and the spouses of the authors for their understanding during the course of this study.

## Author Contributions

**Conceptualization:** Sunday Charles Adeyemo, Oladunni Opeyemi, Calistus Akinleye.

**Data curation:** Sunday Charles Adeyemo, Abdulwaris Salisu Maleka.

**Formal analysis:** Sunday Charles Adeyemo, Funso Olagunju, Eniola Dorcas Olabode.

**Methodology:** Gbadebo Jimoh Oyedeji.

**Supervision:** James Atolagbe, Oladunni Opeyemi.

**Validation:** Sunday Olarewaju.

**Writing – original draft:** Eniola Dorcas Olabode.

**Writing – review & editing:** Sunday Charles Adeyemo.

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
