## [Decision Letter · Decision Letter 0]

2 Sep 2024

Dear Dr Adeyemo,

Thank you very much for submitting your manuscript "EFFECT OF HEALTH EDUCATION IN THE CONTROL OF SCHISTOSOMIASIS IN DASS EMIRATE COUNCIL OF BAUCHI STATE, NIGERIA: AN INTERVENTION STUDY" for consideration at PLOS Neglected Tropical Diseases. As with all papers reviewed by the journal, your manuscript was reviewed by members of the editorial board and by several independent reviewers. In light of the reviews (below this email), we would like to invite the resubmission of a significantly-revised version that takes into account the reviewers' comments. 

We cannot make any decision about publication until we have seen the revised manuscript and your response to the reviewers' comments. Your revised manuscript is also likely to be sent to reviewers for further evaluation.

Sincerely,

Feng Xue, Ph.D.

Guest Editor

Jong-Yil Chai

Section Editor

Reviewer's Responses to Questions

**Key Review Criteria Required for Acceptance?**

**Methods**

-Are the objectives of the study clearly articulated with a clear testable hypothesis stated?

-Is the study design appropriate to address the stated objectives?

-Is the population clearly described and appropriate for the hypothesis being tested?

-Is the sample size sufficient to ensure adequate power to address the hypothesis being tested?

-Were correct statistical analysis used to support conclusions?

-Are there concerns about ethical or regulatory requirements being met?

Reviewer #1: Somewhat, no sufficient details were given about the intervention.

Reviewer #2: The method and the procedure use to address the aims are acceptable.

Reviewer #3: Needs major revision

Reviewer #4: The objectives of the study are clearly articulated with a clear testable hypothesis stated.

The study design appropriate is well to address the stated objectives.

The population is clearly described and appropriate for the hypothesis being tested.

The sample size is more than sufficient to ensure adequate power to address the hypothesis being tested.

The statistical analysis used to support conclusions is done correctly.

There are no concerns about ethical or regulatory requirements being met,

**Results**

-Does the analysis presented match the analysis plan?

-Are the results clearly and completely presented?

-Are the figures (Tables, Images) of sufficient quality for clarity?

Reviewer #1: yes

Reviewer #2: The pre-intervention data reveals a high prevalence of schistosomiasis and generally low levels of knowledge and preventive practices, which aligns with the study’s context. The manuscript effectively highlights the significant improvements in knowledge and reductions in schistosomiasis prevalence post-intervention. The discussion appropriately interprets the findings in the context of existing literature, particularly the significant reduction in schistosomiasis prevalence and the increase in community knowledge. The manuscript does well to link these outcomes to the health education intervention. The quality of the figure need to be improved.

Reviewer #3: The Statistical Analysis needs major revision and clarity.

Reviewer #4: The analysis presented match the analysis plan.

The results are indeed clearly and completely presented.

The figures (Tables, Images) are of sufficient quality for clarity.

**Conclusions**

-Are the conclusions supported by the data presented?

-Are the limitations of analysis clearly described?

-Do the authors discuss how these data can be helpful to advance our understanding of the topic under study?

-Is public health relevance addressed?

Reviewer #1: yes

Reviewer #2: The recommendation for integrating health education into broader schistosomiasis control programs is particularly appreciated and the conclusion effectively summarizes the study's key findings and their implications. It is concise and ties back to the study's objectives.

Reviewer #3: needs major revision

Reviewer #4: The conclusions are supported by the data presented.

All the limitations of analysis are clearly described.

The authors discuss how these data can be helpful to advance our understanding of the topic under study.

Public health relevance is well addressed

**Editorial and Data Presentation Modifications?**

Reviewer #1: Major revision:

A brief description of the health education programme implemented should be added. 

Suggestions for minor revisions :

31: a major

34: Hence,

38: the majority

46: attitudes ( repeated correction )

76: a training

77: including,

79: is mostly

80: questionnaires were

96: the majority

92: schistosomiasis is the name of the disease, not Genus; it should not be capitalized ( repeated 

 97: healthcare

105: food,

106: preventable,

111: practices, can=could, the use 

123: insect treated = insecticide-treated nets

132: take responsibility for taking the preventive measures

Reviewer #2: (No Response)

Reviewer #3: major revision

Reviewer #4: NA

**Summary and General Comments**

Reviewer #1: The plan for the training should be given e.g. contents, duration staffing attendance and estimated costs and comment on feasibility of application in other endemic areas. . This is a major revision.

Reviewer #2: The manuscript presents a well-conducted intervention study with clear and significant findings regarding the impact of health education on schistosomiasis control in a high-prevalence area. The strengths of the study include its relevance to public health, the use of a quasi-experimental design, and the significant improvements observed in community knowledge and disease prevalence. However, the manuscript could benefit from additional details on the intervention itself, a more thorough discussion of the findings, and a more robust consideration of the study’s limitations. Overall, the study makes a valuable contribution to the field of parasitic disease control and provides actionable insights for public health practitioners. Additionally, the manuscript and the study present some weaknesses that need to be clearly addressed by the authors

Reviewer #3: (No Response)

Reviewer #4: NA

PLOS authors have the option to publish the peer review history of their article (what does this mean?). If published, this will include your full peer review and any attached files.

Reviewer #1: Yes: Ahmed Adeel

Reviewer #2: Yes: Ulrich Femoe Membe

Reviewer #3: Yes: Manfred Dakorah Asiedu

Reviewer #4: Yes: Ana Júlia Pinto Fonseca Sieuve Afonso, Universidade Nova de Lisboa, Portugal.
---

## [Decision Letter · Decision Letter 1]

25 Oct 2024

PNTD-D-24-00915R1EFFECT OF HEALTH EDUCATION IN THE CONTROL OF SCHISTOSOMIASIS IN DASS EMIRATE COUNCIL OF BAUCHI STATE, NIGERIA: AN INTERVENTION STUDYPLOS Neglected Tropical Diseases Dear Dr. Adeyemo, Thank you for submitting your manuscript to PLOS Neglected Tropical Diseases. After careful consideration, we feel that it has merit but does not fully meet PLOS Neglected Tropical Diseases's publication criteria as it currently stands. Therefore, we invite you to submit a revised version of the manuscript that addresses the points raised during the review process. Please submit your revised manuscript within 30 days Nov 24 2024 11:59PM. If you will need more time than this to complete your revisions, please reply to this message or contact the journal office at plosntds@plos.org. Please include the following items when submitting your revised manuscript:*
A rebuttal letter that responds to each point raised by the editor and reviewer(s). You should upload this letter as a separate file labeled 'Response to Reviewers'. This file does not need to include responses to any formatting updates and technical items listed in the 'Journal Requirements' section below.*
A marked-up copy of your manuscript that highlights changes made to the original version. You should upload this as a separate file labeled 'Revised Manuscript with Track Changes'.*
An unmarked version of your revised paper without tracked changes. You should upload this as a separate file labeled 'Manuscript'. If you would like to make changes to your financial disclosure, competing interests statement, or data availability statement, please make these updates within the submission form at the time of resubmission. Guidelines for resubmitting your figure files are available below the reviewer comments at the end of this letter. We look forward to receiving your revised manuscript. Kind regards, Feng Xue, Ph.D.Guest EditorPLOS Neglected Tropical Diseases Jong-Yil ChaiSection EditorPLOS Neglected Tropical Diseases

Shaden Kamhawi

co-Editor-in-Chief

Paul Brindley

co-Editor-in-Chief

 **Journal Requirements:** **Additional Editor Comments (if provided):****Reviewers' comments:** Reviewer's Responses to Questions

**Key Review Criteria Required for Acceptance?**

**Methods**

-Are the objectives of the study clearly articulated with a clear testable hypothesis stated?

-Is the study design appropriate to address the stated objectives?

-Is the population clearly described and appropriate for the hypothesis being tested?

-Is the sample size sufficient to ensure adequate power to address the hypothesis being tested?

-Were correct statistical analysis used to support conclusions?

-Are there concerns about ethical or regulatory requirements being met?

Reviewer #1: Yes

Reviewer #2: The objective of this study are clearly mentionned within the manuscript.

Reviewer #3: Clarity and Structure: The structure is generally clear, but the flow could be improved. The description of the sampling techniques and intervention processes is detailed but occasionally overwhelming due to the amount of technical information provided in a single paragraph. Breaking down the sampling method into distinct sections (e.g., Stage 1, Stage 2, Stage 3) with subheadings could improve readability.

Bias and Limitations: The study does not discuss potential sources of bias (e.g., selection bias, response bias) or limitations related to its quasi-experimental design. Without a control group, it’s difficult to rule out external factors that may have influenced the post-intervention results. This could have been acknowledged and addressed in the methodology section

Reviewer #4: -Are the objectives of the study clearly articulated with a clear testable hypothesis stated? yes they are

-Is the study design appropriate to address the stated objectives? yes it is

-Is the population clearly described and appropriate for the hypothesis being tested? it is well described

-Is the sample size sufficient to ensure adequate power to address the hypothesis being tested? the sample size is appropriate

-Were correct statistical analysis used to support conclusions? yes the statistical anaysis is corretly done

-Are there concerns about ethical or regulatory requirements being met? all ethique and regulatory requirements have been met

**Results**

-Does the analysis presented match the analysis plan?

-Are the results clearly and completely presented?

-Are the figures (Tables, Images) of sufficient quality for clarity?

Reviewer #1: yes

Reviewer #2: The data analysis is well justified. The results are clearly presented and the figure quality was improved.

Reviewer #3: (No Response)

Reviewer #4: -Does the analysis presented match the analysis plan? The analysis match the analysis plan

-Are the results clearly and completely presented? The results are completely presented

-Are the figures (Tables, Images) of sufficient quality for clarity? I would change the colours of the table to a more neutral colour

**Conclusions**

-Are the conclusions supported by the data presented?

-Are the limitations of analysis clearly described?

-Do the authors discuss how these data can be helpful to advance our understanding of the topic under study?

-Is public health relevance addressed?

Reviewer #1: Yes

Reviewer #2: The conclusion is concise and the study liitations are clearly mentionned and improved

Reviewer #3: (No Response)

Reviewer #4: Are the conclusions supported by the data presented? The conclusions are suported by the data presented

-Are the limitations of analysis clearly described? The limitations are clearly described

-Do the authors discuss how these data can be helpful to advance our understanding of the topic under study? The authors discuss how these data can be halpul to advance and understand the topic under study

-Is public health relevance addressed? Public health relevance is fully addressed.

**Editorial and Data Presentation Modifications?**

Reviewer #1: In editing the manuscript, please note that "schistosomiasis" is the name of the disease and should not be capitalized like the taxonomic name of the parasite.

Reviewer #2: My recommandation for this this manuscript would be "Accept"

Reviewer #3: (No Response)

Reviewer #4: Please consider changing the graph colours to a more clear and user friendly colour.

**Summary and General Comments**

Reviewer #1: In this version the authors have addressed our previous concern about lack of data on the intervention.

Reviewer #2: N/A

Reviewer #3: In summary, the methodology section presents a well-organized and comprehensive approach to data collection and intervention delivery. However, it could be improved by providing more details on cultural adaptations, data quality control measures, and addressing potential limitations of the study design.

Reviewer #4: Important paper.

PLOS authors have the option to publish the peer review history of their article (what does this mean?). If published, this will include your full peer review and any attached files.

Reviewer #1: **Yes: **Ahmed A. Adeel

Reviewer #2: No

Reviewer #3: **Yes: **Manfred Dakorah Asiedu

Reviewer #4: **Yes: **Ana Júlia Pinto Fonseca Sieuve Afonso

---

## [Decision Letter · Decision Letter 2]

17 Dec 2024

Dear Dr Adeyemo,

We are pleased to inform you that your manuscript 'EFFECT OF HEALTH EDUCATION IN THE CONTROL OF SCHISTOSOMIASIS IN DASS EMIRATE COUNCIL OF BAUCHI STATE, NIGERIA: AN INTERVENTION STUDY' has been provisionally accepted for publication in PLOS Neglected Tropical Diseases.

Best regards,

Feng Xue, Ph.D.

Guest Editor

Jong-Yil Chai

Section Editor

Shaden Kamhawi

co-Editor-in-Chief

Paul Brindley

co-Editor-in-Chief

Reviewer's Responses to Questions

**Key Review Criteria Required for Acceptance?**

**Methods**

-Are the objectives of the study clearly articulated with a clear testable hypothesis stated?

-Is the study design appropriate to address the stated objectives?

-Is the population clearly described and appropriate for the hypothesis being tested?

-Is the sample size sufficient to ensure adequate power to address the hypothesis being tested?

-Were correct statistical analysis used to support conclusions?

-Are there concerns about ethical or regulatory requirements being met?

Reviewer #1: The objectives of the study clearly articulated appropriate designed, adequate statistical analysis that support conclusions.

Reviewer #2: All my comments have been addressed with satisfaction

Reviewer #3: (No Response)

**Results**

-Does the analysis presented match the analysis plan?

-Are the results clearly and completely presented?

-Are the figures (Tables, Images) of sufficient quality for clarity?

Reviewer #1: yes

Reviewer #2: The results presentation are now acceptable.

Reviewer #3: (No Response)

**Conclusions**

-Are the conclusions supported by the data presented?

-Are the limitations of analysis clearly described?

-Do the authors discuss how these data can be helpful to advance our understanding of the topic under study?

-Is public health relevance addressed?

Reviewer #1: yes

Reviewer #2: The conclusion is now more precise

Reviewer #3: (No Response)

**Editorial and Data Presentation Modifications?**

Reviewer #1: Accept.

Reviewer #2: (No Response)

Reviewer #3: (No Response)

**Summary and General Comments**

Reviewer #1: (No Response)

Reviewer #2: The authors have successfully addressed most of the comments. The Methods, the Results and the discussion section have been substantially improved.

Reviewer #3: (No Response)

PLOS authors have the option to publish the peer review history of their article (what does this mean?). If published, this will include your full peer review and any attached files.

Reviewer #1: **Yes: **Ahmed Adeel

Reviewer #2: **Yes: **Ulrich Femoe Membe

Reviewer #3: **Yes: **Manfred Dakorah Asiedu

---

## [Editor Report · Acceptance letter]

29 Dec 2024

Dear Dr Adeyemo,

We are delighted to inform you that your manuscript, "EFFECT OF HEALTH EDUCATION IN THE CONTROL OF SCHISTOSOMIASIS IN DASS EMIRATE COUNCIL OF BAUCHI STATE, NIGERIA: AN INTERVENTION STUDY," has been formally accepted for publication in PLOS Neglected Tropical Diseases.

Best regards,

Shaden Kamhawi

co-Editor-in-Chief

Paul Brindley

co-Editor-in-Chief
